# Pancreatic Cancer Molecular Classifications: From Bulk Genomics to Single Cell Analysis

**DOI:** 10.3390/ijms21082814

**Published:** 2020-04-17

**Authors:** Luca Pompella, Giuseppe Tirino, Annalisa Pappalardo, Marianna Caterino, Anna Ventriglia, Valeria Nacca, Michele Orditura, Fortunato Ciardiello, Ferdinando De Vita

**Affiliations:** Division of Medical Oncology, Department of Precision Medicine, School of Medicine, University of Campania “L. Vanvitelli”, Via Pansini n. 5, 80131 Naples, Italy; giuseppe.tirino@unicampania.it (G.T.); annalisa.pappalardo88@gmail.com (A.P.); caterinomarianna@gmail.com (M.C.); anna.ventriglia@gmail.com (A.V.); valeria.nacca91@libero.it (V.N.); michele.orditura@unicampania.it (M.O.); fortunato.ciardiello@unicampania.it (F.C.)

**Keywords:** pancreatic cancer, genomics, sc-RNAseq, tumor microenvironment

## Abstract

Pancreatic cancer represents one of the most lethal disease worldwide but still orphan of a molecularly driven therapeutic approach, although many genomic and transcriptomic classifications have been proposed over the years. Clinical heterogeneity is a hallmark of this disease, as different patients show different responses to the same therapeutic regimens. However, genomic analyses revealed quite a homogeneous disease picture, with very common mutations in four genes only (KRAS, TP53, CDKN2A, and SMAD4) and a long tail of other mutated genes, with doubtful pathogenic meaning. Even bulk transcriptomic classifications could not resolve this great heterogeneity, as many informations related to small cell populations within cancer tissue could be lost. At the same time, single cell analysis has emerged as a powerful tool to dissect intratumoral heterogeneity like never before, with possibility of generating a new disease taxonomy at unprecedented molecular resolution. In this review, we summarize the most relevant genomic, bulk and single-cell transcriptomic classifications of pancreatic cancer, and try to understand how novel technologies, like single cell analysis, could lead to novel therapeutic strategies for this highly lethal disease.

## 1. Introduction

Pancreatic cancer (PC) is among the most lethal cancers worldwide, with 432.242 deaths estimated in 2018 [1], and an overall 5-year survival rate of 8% [2], which makes it the seventh leading cause of cancer related death globally [3]. As a matter of fact, it is expected that this disease will become in 2030 the second leading cause of cancer death [4]: therefore, there is an urgent need of novel therapeutic strategies.

Pancreatic ductal adenocarcinoma (PDAC) is the prevalent histological subtype of PC [5,6], representing almost 90%–95% of all pancreatic exocrine tumors.

Although we refer to PDAC as uniformly aggressive disease, we commonly see, in the clinical practice, a great clinical heterogeneity among patients, with different clinical and radiological responses to same chemotherapy regimens.

In stage III-IV disease, excluding rare cases of PDAC with BRCA-deficient or MSI-high tumors (who respond to PARP-inhibitors following platinum based induction [7] or immunotherapy [8], respectively), the majority of patients show an objective response to first line chemotherapy for limited period, before developing disease progression (PD), while others only show stable disease (SD) and very rapidly develop PD. Moreover, some patients do not respond at all, developing PD in the meantime of first-line therapy, with very poor prognostic implications.

As for metastatic patients, we also see great heterogeneity among resected ones (stage I-II): Most of them relapse within few months, and only a minority is really cured, regardless of adjuvant therapy used.

This strong clinical heterogeneity cannot be explained by genomic studies only, as they [9,10,11] continued to show us, over the years, the image of a quite “homogeneous” disease, with recurrent mutations in 4 major genes (KRAS, TP53, CDKN2A, SMAD4) and a constellation of other molecular alterations, of uncertain meaning in most cases. Therefore, if most PDAC originates with similar mutational path, it is a matter of fact that other non-genetic mechanisms (microenvironmental interactions or epigenetic mechanisms) are crucial in defining the clinical disease course and aggressiveness.

One way to dissect aggressiveness and classify PDAC besides genomics is the transcriptomic analysis. Bulk transcriptomic classifications [12,13,14,15] and, more recently, analysis of transcriptome in single-cells with novel techniques like single-cell-RNA-sequencing (scRNA-seq) [16,17,18,19] have been able to identity new subtypes of PDAC (with different clinical behavior) and different cell populations/cell states within PDAC microenvironment, respectively, with strong implications for future development of targeted therapies.

In this review we summarize the most relevant genomic, bulk and single-cell transcriptomic classifications of PDAC and try to understand how novel technologies, like scRNA-seq, could deeply dissect intratumoral heterogeneity giving rise to a novel disease taxonomy and possibly novel targets for immunotherapy or other therapeutic strategies for this awful disease.

## 2. Pancreatic Ductal Adenocarcinoma: from Histology to Early Genomic Studies

PDAC accounts for almost all (> 90–95%) exocrine pancreatic tumors [6,20] and the majority of cases is defined as “not otherwise specified” (NOS), according to current World Health Organization (WHO) classification [6]. PDACs are mainly sporadic cancers (only 10% raise in the context of familial/hereditary cancer [21]) and may arise anywhere in the pancreas, but most commonly in the head (60% to 70%) followed by body (10%) and tail (10–15%) [22]. Histologically, PDAC NOS lesions show a strong desmoplastic reaction, a very peculiar aspect, as many pathologists symbolically refer to them as “scars with only a few cancer cells inside” [5,22]. Indeed, cancerous tubular glands, composed of mucin producing epithelial cells (ranging from mild to severe atypia), are totally incorporated within this striking desmoplasia, that is composed of extensive extracellular matrix proteins. Within this hard fibrous tissue, cancer-associated fibroblasts (CAFs) and other immune cells reside and exchange informations each other and with cancer cells.

The majority of PDACs originates from microscopic precursors called pancreatic intraepithelial neoplasia (PanIN) [23,24], not visible with common imaging techniques, while only a minority arises from pancreatic cystic lesions (intraductal papillary mucinous neoplasms (IPMNs) and mucinous cystic neoplasms), which are conversely detectable with CT or MRI scan [25].

More in detail, PanINs are non-infiltrating microscopic intraductal lesions with diameter < 0.5 cm, composed of cuboid/columnar mucinous cells with varying degree of dysplasia, from low-grade (PanIN 1) to high-grade (PanIN 3), the latter one almost exclusively reported in association with infiltrating PDAC [26]. Early lesions (PanINs 1) show frequent KRAS somatic mutations, while PanINs 3 display also CDKN2A, TP53, and SMAD4 mutations, all of which significantly enriched in fully transformed PDAC (see below), as TP53 and SMAD4 inactivation appear to be the latest events of the molecular cascade that lead to overt PDAC. In fact, isolated PanIN 3 (in the absence of PDAC) do not show TP53 or SMAD4 mutations [27], reinforcing once more the idea that these mutations are only enriched in lesions that evolve to invasive cancers. Regarding IPMNs, a recent analysis [28] identified recurrent mutations in GNAS (almost 80%) as well as in KRAS (50%), with TP53 and SMAD4 gene mutations only observed in high-grade IPMN. These mutations, in fact, could be seen as the latest molecular events also in IPMN carcinogenesis [27], similarly to high-grade PanIN.

As already mentioned, overt PDACs present significantly recurrent mutations (well-known from the late 1990s) in four genes only: gain-of-function mutation in KRAS oncogene; loss-of-function mutations in TP53, CDKN2A and SMAD4 tumor suppressor genes [29,30,31]. Besides these genes, a “long tail” of less prevalent alterations in other protein-coding genes has been identified over the years, with more or less clear pathogenic implications [10,32].

Genomic research in pancreatic cancer has always been very difficult, considering that a typical PDAC often contains only 5% to 20% of neoplastic cells [31], and most studies have focused on tumors with neoplastic cellularity typically greater than 40% [10], excluding cancers with lower cellularity.

Jones S. et al. published in 2008 the first attempt to a global molecular dissection of this malignancy [9]: They analyzed 24 advanced PDACs (ranging from stage IIb to stage IV patients) with next generation sequencing (NGS) technology for structural variants and somatic mutations detection in protein-coding genes. Authors have shown that PDACs contain an average of 63 genetic alterations (considerable less than other cancers like breast cancer, *n* = 101, or colorectal cancer, *n* = 77), most of which are point mutations, and confirmed the frequent homozygous deletions in tumor suppressor genes like TP53, CDKN2A, and SMAD4. The real strength of this paper is to have identified 69 genes, significantly altered in the majority of tumors analyzed, that could be grouped into 12 core-signaling pathways, each of which altered in 67% to 100% of the 24 tumor samples. For more details about the core pathways identified see Table 1 (with comparison with core pathways identified by Bailey et al. [14]—see later in the text ), while for a list of the most mutated genes (and comparison with other genomic studies) see Table 2.

The biological relevance of the identified core pathways was also underlined by transcriptome analysis: The authors showed that the genes related to the twelve identified pathways were the most expressed, and this unequivocally supports the contribution of the pathways themselves to pancreatic cancer pathogenesis.

This paper has given us an even more complicated picture than we could have thought: Even if different PDACs contain mutations in genes involved in most of the 12 pathways identified, the specific genes altered in any individual tumor can widely vary. The authors could not demonstrate how mutations in different genes of the same core pathway could affect pancreatic tumorigenesis in such different way.

The main limits of this early genomic study are the low number of samples analyzed, as well as the search for alteration exclusively in the coding part of the genome, without analysis of regulatory elements. These limitations will be partially overcome by subsequent papers, as we will describe in the next section.

## 3. Genome Sequencing of PDAC: From Exome Sequencing to Whole Genomes and Multi-Omics

In 2012, a large consortium driven by Sean Grimmond published a seminal paper [32] in which 99 high informative PDAC samples, from a clinical cohort of 142 early stage patients (stage I and II PDAC), were analyzed using a combination of capture systems and next generation sequencing platforms, for whole exome sequencing analysis. Patients were all affected by untreated resectable PDAC, which underwent pancreatectomy with curative intent. The average number of mutations detected per patient was 26 (range 1–116): authors confirmed the majority of mutations previously identified by Jones [9], but also defined a large number of novel mutations (1456 genes), mostly occurring at very low frequency, and with doubtful pathogenetic meaning. The most frequently mutated genes were found to be KRAS, TP53, CDKN2A, SMAD4, MLL3, ARID1A, and SF3B1, to which authors added novel significantly mutated genes (like ARID2 and ATM).

The pathway analysis of the altered genes (similarly to Jones et al. [9]) confirmed the clustering into known core pathways (apoptosis, TGF-beta signaling, G1/S checkpoint machinery, etc.), but found a novel gene signature, the so-called “axon-guidance”, which was associated for the first time to PDAC pathogenesis.

Axon-guidance genes (semaphorins, slits, netrins, and ephrins) are fundamental regulators of neuronal migration and positioning during embryogenesis: in the present study, SLIT2 and ROBO2 mutations were found in 5% of patients, with focal copy number losses of ROBO1 and SLIT2 in an additional 15% of the cohort. The clinical significance of these alterations is clarified by the fact that low mRNA expression of ROBO2 receptor was correlated with poor prognosis, assuming the role of SLIT2 as a tumor suppressor gene.

Moreover, amplification of class 3 semaphorins (SEMA3A and SEMA3E) was detected in 18% of patients and mutated in 3%: Again, of clinical interest, high mRNA expression of SEMA3A and PLXNA1 (another molecule crucial in semaphorins pathway) was significantly associated with poor survival, both in univariate and multivariate analysis. This paper [32] greatly highlighted the power of next generation sequencing, applied to a homogeneous clinical cohort, for the discovery of novel genomic pathways, possibly crucial for PDAC pathogenesis, with some implications for the design of novel targeted drugs.

Three years later, a big scientific consortium [10], partially overlapping with the previous work, published the first comprehensive whole genome sequencing (WGS) study of patients affected by PDAC.

The authors performed deep WGS and copy number variation analysis of 100 early stage PDACs: They confirmed the most common mutated genes already known (like KRAS, TP53, CDKN2A, SMAD4), as well as the recently described “Axon-Guidance Pathway” gene alterations (ROBO1, ROBO2, SLIT2) [32], but discovered also novel genes, never before implicated in PDAC pathogenesis (like KDM6A and PREX2). See Table 2 for a comparison between the main mutated genes across the principal genomic studies performed so far.

Moreover, combining structural variations with point mutations, the prevalence of inactivation events for “classical” PDAC tumor suppressor genes like TP53, CDKN2A, and SMAD4 were further increased to 74%, 35%, and 31%, respectively.

The most important finding was the comprehension that chromosomal structural variation is a crucial mechanism of DNA damage in PDAC pathogenesis: in fact, different patterns of structural variation classified PDACs in four subtypes, with potential clinical implications. These subtypes were named “stable”, “locally rearranged”, “scattered”, and “unstable”.

“Stable” subtype (20% of all samples) presents defects in cell cycle/mitosis processes, with high aneuploidy, but it contains ≤ 50 structural events. Point mutations in KRAS, SMAD4 and TP53 were quite similar to the other groups.

“Locally rearranged” subtype (30% of all samples) is so called because of the significant focal events in just one or two chromosomes: One third of these cases presents focal amplifications in known oncogenes (like KRAS, GATA6 and others), some of which (ERBB2, MET, CDK6, PIK3CA) could be targeted with available drugs, although these alterations exist at very low prevalence in the cohort (only 1%–2% of all patients).

“Scattered” subtype (36% of all samples) exhibits non-random chromosomal damages and ≤ 200 structural alteration events.

Of more interest, the “unstable” subtype (14% of all samples) presents serious defects in DNA integrity maintenance, with consequent high genomic instability (> 200 structural variation events). The authors could identify a mutational signature associated with loss-of-function mutation in BRCA pathway (the so called “BRCA signature”) in the majority of “unstable” tumors (10/14). The main genes found to be altered (and causative “agent” of this signature) were BRCA1 (*n* = 2), BRCA2 (*n* = 7) and PALB2 (*n* = 2). A minority of this mutations were inherited (germline mutations), while others were of somatic origin.

The paper has very important clinical implications, since authors showed that among five “unstable” patients (high BRCA signature) treated with platinum-based regimen, two had exceptional radiological (complete response according to RECIST1.1 criteria [34]) and clinical responses, while other two obtained partial responses (according to RECIST1.1). The analysis of these responses was the first evidence ever of a possible predictive biomarker for platinum responsiveness in PDAC. Indeed, the recent positive results of the “POLO” Trial [7], with Olaparib maintenance after platinum induction therapy in germinal BRCA1/2 mutated PDAC patients, were in fact all built on the proof-of-concept data presented here [10].

The transition from genomic characterization only to multi-omic analysis of PDAC was short: just two years later, in 2017, The Cancer Genome Atlas (TCGA) Research Network (lead by Raphael BJ) [11] published a seminal paper in which 150 PDAC samples (stage I-III patients) were analyzed through genomic (whole exome sequencing), transcriptomic (RNA sequencing) and proteomic profiling.

Again, only patients with resectable (and de facto resected) disease were enrolled, as for the Jones [9] and Waddell [10] studies. Whole exome sequencing confirmed the high mutation rate within the “usual suspects” (KRAS, TP53, CDKN2A, SMAD4) and, at lower levels, in RNF43, ARID1A, TGFBR2 and GNAS (see Table 2), already descripted by previous researchers. The only gene not previously reported as mutated in PDAC was RREB1, which has presumably an important role for zinc homeostasis in PDAC pathophysiology. Moreover, almost 8% of the patients included in TCGA cohort presented germline mutations: Six in BRCA2, three in ATM, one in PALB2 and one in PRSS1 (data quite similar to that of Waddell et al. [10]); of note, the majority of these germline alterations was enriched in KRAS wild-type samples (10/11). Concerning to copy number aberrations, the authors observed amplification of GATA6, ERBB2, KRAS, AKT2, and MYC, as well as deletions of CDKN2A, SMAD4, ARID1A, and PTEN.

Interestingly, as already mentioned, some cases (*n* = 10) do not have KRAS mutation: They present mainly somatic genetic alterations that activate in an alternative way the RAS-MAPK pathway upstream or downstream of KRAS itself. For example, mutation of BRAF (*n* = 3) or FGFR4 (*n* = 1), amplification of ERBB2 (*n* = 1) and NF1 (*n* = 1) were the most frequent alterations. Alternative pathways were genetically activated in tumors without RAS-MAPK activation: missense mutation of GNAS gene (*n* = 3), a well-known oncogene in different cancers [35], mainly ocular melanoma, and mutations in CTNNB1 (*n* = 2). To complicate things even more, a recent paper by Glimm et al. [36] identified in KRAS wild type patients recurrent fusions in genes like NRG1 (encoding a ligand for ERBB3 pseudokinase) or RET (encoding a tyrosine kinase receptor), with crucial implications for targeted therapy. In fact, two NRG-rearranged PDAC patients were treated with EGF-R inhibitor (afatinib) or EGF-R inhibitor plus ERBB2 dimerization inhibitor (erlotinib plus pertuzumab) with encouraging results. An additional kinase gene rearranged in KRAS wild type tumors is NTRK: O’Reilly EM [37] showed that one metastatic patient, whose tumor was CTRC-NTRK1 rearranged, received substantial benefit from tropomyosin receptor kinase (TRK) inhibitor larotrectinib, suggesting once more that these KRAS wilt type patients should be identified absolutely, because they could greatly benefit from targeted therapeutic approaches.

Returning to TCGA data, based on the different micro-RNAs (miRNAs) expression, the authors identified three different “miR” tumor clusters, of which miR Cluster 2 seems to be particularly related to RNF43 mutation, with some biological and clinical relevance [38].

To summarize, deep sequencing of PDACs identified some patients (42%) that had a cancer harboring at least one druggable alteration, as well as almost 8% of patients with inherited BRCA pathway alteration that could be treated with platinum and PARP inhibitor [7].

All genomic studies reviewed so far are focused on early stage disease: however, resectable PDACs are ~20% of all pancreatic cancers clinically evident, because the majority of patients is diagnosed in stage IV [30]. In 2017, Iacobuzio-Donahue et al. [39] examined 26 metastases from four PDAC patients with whole genome sequencing technologies: They showed a strong concordance between each primary tumor and its metastases in all driver gene mutations found. Instead, intratumoral heterogeneity (between primary and metastases) seemed to refer exclusively to passenger mutations (that is, mutations without known functional consequences). This work highlighted for the first time the notion that driver mutations in PDAC seem to be all retained during neoplastic progression from primitive to metastasis: Importantly, this concept could have strong clinical implications, since molecular profiling of the primary lesion alone may identify a druggable target that is the same for most of the metastatic lesions too.

The last study we review in this section is from Connor AA et al. [33]: Unlike the previous extensive analyses [9,10,11,32], which focused on early stage disease only, it examined both primary and metastatic lesions of hundreds of patients. Indeed, the authors studied 289 PDAC patients with WGS and RNA-sequencing (RNA-seq), for a total of 319 cancer samples (224 from primary site and 95 from metastatic sites). For 19 patients there were paired samples (primary plus metastasis).

Once again, the overall mutational analysis of the entire cohort confirmed the very high prevalence of KRAS (89% of samples), TP53 (80% of samples), CDKN2A (26% of samples), and SMAD4 (25% of samples) mutations. Other genes recurrently mutated were ARID1A (9%), KDM6A (5%), RNF43 (5%), and others (see Table 2 for a complete picture and comparison with other studies).

No mutation is found to be more frequent in primary than in metastases (in agreement with Iacobuzio-Donahue et al. [39]) and the majority of oncogene mutations were of activating type (gain-of-function): This was true for KRAS (247/247), GNAS (11/11), and BRAF (6/8), as well as for others. Moreover, almost 13% of patients (36 individuals) presented PDAC germline predisposition (the majority had germline mutations in BRCA2 and BRCA1), similarly to what TCGA consortium reported [11].

Of note, the genetic analysis of three paired cancer lesions (primary tumor, a peri-pancreatic lymph node, and one liver metastasis) from the same patient showed a very high concordance, in terms of mutations, between primary and loco-regional lymph node, as well as between the lymph node and the metastatic lesion, but a low concordance between the primary and the liver metastasis. This supports, also in PDAC, as for prostate [40] and ovarian cancer [41], the concept of a gradual tumor progression process, with cancer dissemination to loco-regional lymph nodes first, and only later from the latter ones to liver or other organs.

The transcriptomic analysis showed that at least half of PDACs analyzed present a hypoxic gene signature, both primaries and metastatic lesions, with a high concordance between them. Probably, hypoxia is an intrinsic characteristic of PDAC biology, rather than a mere consequence of the desmoplastic tumor microenvironment. Basal-like [13] and squamous subtype [14] tumors are more hypoxic (high hypoxia signature) than classical like, and they are associated to poor prognosis. For detailed description of transcriptomic subtypes (basal-like, squamous, etc), see the next section.

The work by Connor [33] was the first large study that investigated genomically both primary and metastatic PDAC lesions, reaffirming strongly the concept (already emerged from Iacobuzio-Donhaue paper [39]) that PDAC seems quite homogeneous between primary and secondary lesions, with similar mutations operated by similar mutational processes in different tumor locations. In conclusion, although most of the genomic studies reviewed so far showed a moderate concordance between the different driver genes identified (see Table 2), some differences also emerged, because of the different distribution of disease stage (early versus advanced), grading (low versus high) or other clinical/pathological characteristics between the different patients enrolled in the various studies themselves.

## 4. Bulk Transcriptomic Sub-Typing Paves the Way to Dissect Different Clinical Behavior as well as Different Drug Responses in PDAC

Analysis of the cancer transcriptome could be very useful to define intrinsic molecular subtype of a particular tumor, with different biological, clinical and prognostic implications. To demonstrate the great power of transcriptomic, even when it was still in its infancy, we like to refer as an example to a fundamental study [42], conducted in early 2000s in lymphomas, with a technology called DNA microarray. Alizadeh et al. showed, for the first time, that Diffuse Large B-Cell Lymphoma (DLBCL), previously thought to be “a single disease”, was in reality a set of at least three different entities (depending on differentiation state of lymphoma cells) with different responses to the same R-CHOP chemotherapy regimen and very different prognosis.

The first major transcriptomic work in PDAC was by Collisson et al. [12]: in 2011 they analyze 27 resected pancreatic tumors with gene-expression microarray and identified a 62-gene signature (called “PDAssigner gene set”) able to distinguish three different subtype of PDAC (Classical type, Quasimesenchymal type and Exocrine-like). PDAssigner was then applied to 19 PDAC human cell lines and 15 PDAC murine cell lines, with some very interesting findings. See Table 3 for transcriptomic subtypes comparison from different studies.

Classical type showed the highest expression of epithelial and adhesion associated genes, as well as very high mRNA expression of GATA binding protein 6 (GATA6) and KRAS. RNA interference knockdown of GATA6 in classical cell lines resulted in strong reduction of anchorage independent growth, reinforcing the idea of GATA6 as an essential gene in this subtype. Moreover, in vitro toxicity data showed a high response to Erlotinib but low or no response to gemcitabine in classical cell lines. More importantly, from a clinical point of view, in univariate and multivariate analysis of patient cohort, Classical Subtype resulted as an independent predictor of better survival, compared to Quasimesenchymal Type.

Quasimesenchymal type (QM Type) was so called because of the highest expression of mesenchyme-associated genes. The analysis of QM type cell lines showed high in vitro response to gemcitabine (compared to Classical types) and low or no expression of GATA6. Patients identified as QM type had worse overall survival: This data was confirmed by univariate and multivariate analysis.

Finally, exocrine-like type showed the highest expression of digestive enzyme genes (like ELA3A or CFTR). Of note, the analysis of human and murine PDAC cell lines with “62-gene PDAssigner” was not able to identify even a single exocrine like cell line: Therefore, this subtype is probably an artifact from adjacent normal pancreas contaminating samples, as Moffitt [13] and Puleo [15] will show later.

This study, for the first time ever, could classify PDAC patients in different subtypes, with important implications in terms of biology, prognosis, therapeutic targets and pharmacologic response to available drugs (like gemcitabine or erlotinib). Collisson et al., de facto, pave the way to a more modern concept of cancer classification, towards precision medicine.

Four years later, Moffitt and colleagues [13] performed a virtual microdissection of 141 primary and 61 metastatic PDAC samples and analyzed them with both DNA microarray and RNA-sequencing (RNAseq), although the latter one was applied to a small percentage of cases. They discovered at least four subtypes, two stroma-related (“normal type” and “activated type”) and two cancer-related (“classical” and “basal-like”), (Table 3).

“Normal” stromal subtype showed high expression of pancreatic stellate cell markers (ACTA2, VIM, DES) and was related to a good prognosis (median overall survival (mOS) 24 months, with 1-year survival rate of 82%).

Conversely, “activated” stromal subtype highly expressed macrophage-related genes (integrins like ITGAM, and chemokines like CCL13 and CCL18), tumor promoting genes (SPARC, WNT2, WNT5a, MMP9, and MMP11), and Fibroblast activation protein (FAP), indicating a strong activated fibroblast state. This subtype showed the worse prognosis: mOS 15 months and 1-year survival rate of 60%.

Regarding to tumor-specific subtypes, the Moffitt “classical” type exactly matches with Collisson’s one [12]: It highly expresses GATA6 (indicative of epithelial cell differentiation) and has the best prognosis (mOS of 19 months). The basal-like subtype, overlapping with other basal-like types in distinct cancers like breast tumors [44], has the worst prognosis (mOS of 11 months). Of note, the prognostic effect of stromal-related and tumoral-related subtypes is cumulative: This implies that a basal-like subtype with an activated stromal signature has the worst prognosis ever.

In 2016 Martinelli et al. [45] studied the relationship between a specific transcriptomic subtype and clinical response to adjuvant therapy in PDAC patients: using the large dataset of ESPAC-3 Trial [46], they classified patients by level of expression of GATA6 gene, a well-known marker of Collisson and Moffitt “Classical” subtypes. Individuals with high GATA6 expression (bona fide Classical Type) received greater benefit from adjuvant 5-Fluorouracil (5-FU) while patients with medium or low GATA6 expression (bona fide Basal-like tumors) did not benefit from adjuvant 5-FU. Unfortunately, no relationship between GATA6 subtypes and adjuvant gemcitabine response was found. These data greatly highlighted the importance of a precise molecular dissection (in this case with a single simple marker) to choose the best possible therapy for each patient, even in an adjuvant context.

In the same year, Bailey et al. [14] analyzed 456 resected pancreatic tumors (PDACs with all their variants: Adenosquamous, IPMN-associated PDAC, colloid PDAC, etc) with WGS/WES as well as RNAseq: They identified four different PDAC subtype (Table 3), each associated with a peculiar transcriptional network. Moreover, thirty-two genes were found to be recurrently mutated in a statistically significant way: They were all grouped in ten different core pathways (see Table 1 for a direct comparison with those of Jones et al. [9]).

Using bulk RNAseq applied to 96 tumors with high epithelial component (>40%), the authors described four stable transcriptional classes (Squamous, Pancreatic Progenitor, Immunogenic, and Aberrantly Differentiated endocrine–exocrine), subsequently confirmed with array based mRNA expression on additional 232 PDACs.

Squamous subtype shows strong upregulation of TP63 gene, as well as upregulation of other genes highly expressed by the “squamous-like” classes of breast, bladder, and lung cancers [47]. Most of adenosquamous PDAC present this particular signature, with high TGF-β signaling, MYC pathway activation and integrin α_6_β_4_ signaling. These tumors strongly downregulate genes like GATA6, crucial in the “pancreatic endodermal cell fate determination”, with a complete loss of endodermal identity. Poor prognosis is a hallmark of squamous subtype: in fact, it correlates with Collisson Mesenchymal [12] and Moffitt Basal-like [13] molecular classes.

Pancreatic Progenitor subtype expresses transcriptional networks involving PDX1, FOXA2, and FOXA3 genes, as well as other transcription factors, all of them crucial for the pancreatic cell fate determination. This subtype is the ideal opposite to the squamous class, as it corresponds to Classic-type of Collisson. Moreover, different apomucins (MUC5AC and MUC1) are highly expressed in these cancers, and they fall within the IPMN-associated PDAC class.

Aberrantly-Differentiated Endocrine–Exocrine (ADEX) subtype could be viewed as a sub-class of Pancreatic Progenitor: it expresses transcriptional networks crucial for more advanced stage of normal pancreatic differentiation. Ideally, this subtype corresponds to Exocrine-like of Collisson, although TCGA study [11] and Puleo work [15] later would have shown that both ADEX and Exocrine-like probably appear due to low tumor purity, as consequences of normal pancreas contamination.

Finally, Immunogenic Subtype correlates with strong immune infiltrate: It shows high expression of at least nine different immune cells (mainly B cells and T cells, both CD4+ and CD8+), with upregulation of immune checkpoints CTLA4 and PD1, suggesting a possible role for immunotherapy drugs (like anti-CTLA4 and anti-PD1/PDL1) in this PDAC subclass.

In 2018, Puleo et al. [15] analyzed the genome and transcriptome of 309 resected PDAC tissues, confirming the reliability of previously reported Basal-Like and Classical tumor subtypes. Moreover, they classified PDACs in five robust subtypes (Table 3). Three were mainly defined by tumoral component (“pure classic”, “immune classic”, and “pure basal-like”) and two were greatly influenced by microenvironment and stroma (“desmoplastic” and “stroma activated”).

The two identified classic subtypes, mainly well differentiated, were all composed of classical samples according to Moffitt [14]: One was characterized by low immune infiltration (“pure classic”) with no immune checkpoint upregulation, and the other one was characterized by strong stromal signature (so called “immune classic”). More in detail, the immune classic subtype was highly infiltrated by natural killer, T and B cells, with the highest CTLA4 expression of all other subtypes, making it potentially sensitive to immune checkpoint inhibitors like ipilimumab.

The pure basal-like subtype was exclusively composed of Moffitt basal-like samples, with low immune infiltration: it was predominantly composed of poor differentiated tumors, more often CDKN2A or TP53 mutated, and it exhibited the worst outcome (median overall survival of 10 months versus 43.1 and 37.4 months of pure classical and immune classical types, respectively). Of note, this subtype upregulated B7-H3 and TIM3 inhibitory checkpoints, with some therapeutic potentials.

The other two subtypes were a mixture of classical and basal-like tumors, whose characteristics were dictated by stromal microenvironment composition. Tumors enriched in activated stroma component (high α-SMA, SPARC, FAP) were defined “stroma activated”, while tumors with low tumoral component and a massive stromal and immune infiltration (particularly structural and vascularized stromal components) were named desmoplastic. From a prognostic point of view, a high stromal signature (both of “activated type” or “desmoplastic”) defined poor prognosis in “classical” tumors, while the same in “pure basal-like” patients caused a positive impact on prognosis. For the first time, a different microenvironmental signature is associated to different clinical outcome, depending on the precise subtyping of tumor cell transcriptome.

Indeed, bulk RNAseq cannot resolve the enormous complexity of PDAC microenvironment: supposedly, only analysis of transcriptome in single cells could allow us to understand in deep the pathophysiology of pancreatic cancer, as we will explain in the next section.

## 5. Single-Cell Analysis Reveals the Cellular Ecosystem of PDAC

Recent advances in single cell genomics could provide very powerful tools for an “in-deep analysis” of tumor heterogeneity at unprecedented molecular resolution [48,49]. More in detail, scRNA-seq can characterize in unbiased and systematic manner every single tumoral/stromal cell present in a tissue, analyzing their transcriptomes one by one [50]. With this technology, multiple cell states/cellular subpopulations (and relative cross talks) have been already successfully recognized within the tumor microenvironment in different tumor types [51,52,53,54,55,56,57], with relevant biological and clinical implications.

In 2018, Bernard V et al. [16] applied for the first time scRNA-seq to pancreatic cancer: They analyzed resected samples from six patients (2 low-grade IPMNs, 2 high-grade IMPNs and 2 fully transformed PDACs), with the aim of identifying the microenvironmental and neoplastic cell transcriptional changes that are associated to tumor progression mechanisms. A total of 5403 single cells were sequenced, but for the subsequent analysis 3343 single cells only were considered, due to low quality of gene expression (<300 genes) in 2060 cells. Unsupervised clustering by t-distribution stochastic neighbor embedding (t-SNE) method [58] identified ten different cell clusters, each composed of unique epithelial or stromal/immune element.

To note, most of epithelial cells deriving from low-grade IPMNs (clusters LG.Ep1 and LG.Ep2) were non-proliferating (G_0_ phase), but a small percentage of these (cluster LG.Ep3) showed high expression of oncogenes and cell cycle related genes. That underlines how, also in low-grade lesions, rare cells with higher malignant potential do exist, and they are probably responsible for the neoplastic progression process. Accordingly, in adenocarcinoma-derived clusters (named “PDAC1” and “PDAC2”), were also found some cells, already fully transformed, emerging from low-grade and high-grade IPMN samples (almost 1%–2%). This notion strongly supports the use of single cell transcriptomic tools to identify these small cell populations, because with bulk analysis this data would have been unequivocally lost.

Regarding microenvironmental changes occurring during tumor progression (from low-grade to high grade IPMN to full PDAC), the authors performed a single cell digital microdissection of the entire dataset for stromal/immune cells only, and were able to identify at least seven unique clusters with different cell type distribution in different lesion types.

Low-grade and high-grade IPMNs showed the highest concentration of CD4^+^ and CD8^+^ T cells, while PDAC lesions presented very low levels of these lymphocytes. Indeed, the majority of stromal component in PDAC is made by pro-tumoral Myeloid Derived Suppressor Cells (MDSCs), almost 51% (277/544) of stromal profiled cells in cancer lesions, whereas these cells are very rare in low-grade IPMNs (2.3%: 3/131) and high-grade IPMNs (3.5%: 10/281).

Finally, cancer associated fibroblasts (CAFs) seem to be mainly derived from adenocarcinoma samples. Inflammatory CAFs (iCAFs)—see later in this section for a more complete description—highly express IL6, CXCL12, FAP, and VIM, with very low levels of α-SMA: They form a pro-tumoral population exclusively present in PDAC1 and PDAC2 clusters (10.5% of single stromal cells profiled) and facilitate invasion and metastasis, as well as immune suppression. Myofibroblasts (myCAFs), on the other side, highly express α-SMA and they are probably involved in stromal and endothelial growth factors secretion: This population appears earlier than iCAFs in the cancer evolution process, as it is greatly represented in high-grade IPMNs.

To summarize, in this emerging malignant progression model, there is a tremendous shift in microenvironmental cell populations over time, with gradual depletion of T lymphocytes from low-grade to high-grade IPMN to overt PDAC, and a simultaneous great increase of CAFs (myCAFs in high-grade IPMN and iCAFs in PDAC) and immune suppressive MDSCs in overt cancer. This picture describes a terribly hostile microenvironment to anti-tumor immune system in PDAC and it could explain, at least partially, the very poor prognosis of this cancer.

One year later, Peng J et al. [17] realized the most massive single cell transcriptomic atlas of pancreatic cancer to date, allowing us to take a first look at the entire cellular ecosystem of this disease. The authors analyzed 41.986 single cells from 24 resected PDAC samples and 15.544 single cells from 11 control pancreases: They used principle component analysis on variably expressed genes across all cells from tumoral samples and found 10 main cell clusters.

In detail, they identified two ductal cell clusters: Ductal Type 1 and Ductal Type 2. Both clusters highly express ductal cell markers (like KRT19, MMP7, TSPAN8, and SOX9), but Ductal Type 2, that is the single most represented cluster in the whole dataset (26.9%: 11315/41986 cells), shows a much higher expression of PDAC poor prognosis markers, like KRT19, CECAM1/5/6. More importantly, chromosomal copy number alterations were only found in Ductal Type 2 cells, and this cluster was not recognized in control pancreases. All these data strongly indicate that Ductal Type 2 Cluster is almost exclusively composed of malignant cells: Indeed these cells show upregulation, compared to Ductal Type 1, of more than 3000 genes, all of which are involved in cancer-specific functions (cell proliferation, migration, and hypoxia). On the contrary, Ductal Type 1 cells upregulate genes related to normal pancreatic processes (digestion, pancreatic secretion, bicarbonates transport etc.) and they could be viewed as almost normal cells, although quite different from the Ductal Type 1 cells identified in control pancreases.

More importantly, t-SNE analysis applied to Ductal type 2 cluster identified at least seven different sub-clusters within this cell category, greatly expanding the previously imagined concept of intratumoral heterogeneity (see Figure 1 for a detailed description of each sub-cluster). Sub-clusters 1-2-3 upregulate genes involved in cell differentiation, pancreatic secretion and mRNA translation (all normal pancreatic functions): hence, they could be viewed as “PanIN cell states”, not yet fully transformed. On the other side, sub-clusters 4-5-6-7 show upregulation of several genes involved in different malignant processes. Sub-groups 4–6 upregulate genes related to cell adhesion and migration pathways (thus, they are presumably involved in metastasis), while sub-cluster 5 is involved in immune cell regulation, and sub-cluster 7 (the only one present in all 24 patients) highly upregulates cell cycle genes (thus, it is presumably involved in cancer progression). The latter sub-clusters could be viewed as “PDAC cell states”, composed of fully transformed malignant cells with different functions in terms of microenvironmental modulation properties and metastatic potential.

Highly proliferating ductal cells (mainly sub-cluster 7) identified with scRNA-seq express cell cycle markers like MKI67, CDK1, CCNB1, CCNB2, and others: The authors showed that in vitro inhibition of CDK1, PLK1, and AURKA with specific drugs could significantly reduce malignant ductal cell survival, opening the way to a more targeted therapeutic approach for at least a subgroup of pancreatic cancers. Moreover, they showed that immune T cell response in PDAC samples with high expression of proliferative markers was strongly reduced, although it is not known or even speculated the cause of this inverse relationship.

The authors analyzed also the immune ecosystem in PDAC, mostly focusing on T cells and macrophages, discovering at least five sub-clusters of T lymphocytes (two CD8^+^ and three CD4^+^) and five sub-clusters of macrophages. Regarding T cells, both CD8^+^ sub-clusters strongly express effector T cell genes (GZMA and GZMH), and one (C1-CD8) is highly proliferative; conversely, within CD4^+^ cells there are naïve T cells (C2-CD4), Central Memory T cells (C4-CD4) and T-reg cells (C3-CD4). The clusterization of macrophages reveals five subtypes: sub-cluster 3 is the most abundant and it highly expresses CCL2 chemokine, which is fundamental in MDSCs recruitment, a cell type already proven [16] to be crucial for the establishment of immune suppressive PDAC microenvironment. Other macrophage sub-clusters are involved in antigen presentation (cluster 1), extracellular matrix remodeling (cluster 2), cytotoxic T cells recruitment (cluster 4) and pro-inflammatory cytokine production (cluster 5). In light of this enormous amount of data, it is a matter of fact that the traditional division of macrophages between polarized M1 and M2 types loses all of its old importance.

Unfortunately, the single cell microenvironmental analysis here performed is not complete as that of cancer cells themselves: most of the stromal/immune cells others than T cells or macrophages were absolutely shelved and not described in detail. For example, fibroblasts and CAFs, the most relevant cell type shaping PDAC microenvironment, were not discussed at all and this is, in our opinion, the main limit of this otherwise extraordinary work.

A comprehensive CAFs single cell characterization of human and murine PDAC was the primary objective of Tuveson D [18], whose research group has been involved over many years in CAFs analysis in pancreatic cancer [59,60], as we briefly review before describing in detail his team single cell paper.

In 2017, they proved for the first time the existence of at least two different CAF phenotypes in mouse and human PDAC tissues [59]: Myofibroblasts (myCAFs) and inflammatory CAFs (iCAFs). These two populations activate different signaling pathways and are located in different tumor locations: myCAFs reside in close contact with cancer cells while iCAFs are located far away from neoplastic cells, in tumor desmoplastic areas.

More in detail, myCAFs (so called because of the high levels of α-SMA) are highly contractile cells involved in stroma remodeling (upregulating ACTA2 and TGF-β pathways): Their formation appears dependent on close contact with cancer cells, probably due to juxtracrine interactions. ICAFs, on the other side, are located more distantly from cancer cells within dense tumor stroma: They are activated by paracrine factors secreted from cancer cells themselves and show a strong secretory phenotype. Indeed, iCAF secrete many pro-inflammatory cytokines, like IL-6 and IL-11, which stimulate pro-survival signaling pathways in cancer cells but are also responsible for systemic effects in patients, like cachexia and immune suppression. Importantly, it seems that these two populations can dynamically differentiate from one cell state to the other: The precise mechanisms of this transition has been clarified by the same research group only two years later [60]. See Figure 2 for molecular details.

In 2019, Biffi et al. [60] showed that pancreatic cancer cells could secrete a great amount of both TGF-β and IL1 cytokines: TGF-β strongly activates TGF-β signaling in myCAFs adjacent to tumor cells preventing the induction of iCAF phenotype (by suppressing IL1R1 expression). On the other side, tumor-secreted IL1 activates IL1 signaling in CAFs that are located farther away from tumor cells: in these CAFs, IL1 leads to a strong JAK/STAT signaling activation, which is responsible for the establishment of the iCAF secretory phenotype, and a positive feedback loop on IL1R1, the receptor for IL-1 (Figure 2). Therefore, these two CAF populations seem to be even more interconvertible cell states, rather than terminally differentiated cells, depending on their location within tumor microenvironment. Moreover, JAK inhibition in PDAC mouse models could rapidly shift iCAF to a Myofibroblastic phenotype, supporting the idea of microenvironmental directed drugs as a new therapeutic possibility in PDAC.

Just few months later, Elyada et al. [18] sequenced almost 21.200 single cells from 6 human PDAC patients and 11.260 single cells from 4 KPC PDAC mouse models, with the aim of investigating CAFs heterogeneity at single cell resolution. Density-based clustering of human data identified 15 cell clusters: Three ductal cell clusters and many immune/stromal related clusters. Indeed 89% of single profiled cells were composed of immune cells: B cells, Myeloid cells, T/NK cells, plasma cells, plasmacytoid dendritic cells, mast-cells, and others.

Within ductal cell population, a separated clustering analysis identified at least four different sub-clusters: Cluster 1 and 4 express genes of “classic PDAC type” signature (according to Collisson [12] and Moffitt [13]), thus they are called “Classic 1” and “Classic 2” respectively. Cluster 2 mainly expresses genes involved in lipid metabolism: it seems to be derived from normal pancreatic cells and it is called “lipid processing”. Instead cluster 3 is named “secretory” because it expresses acinar protein related genes. To conclude, Cluster 1-3-4 are composed of tumor cells: In fact, they are enriched in KRAS, TP53, hypoxia and inflammatory pathways, accordingly to inflammatory nature of PDAC. However, basal-like signature genes were not identified in these ductal cells, perhaps because of their low total number.

Regarding immune cells, as in Peng work [17], scRNA-seq identified different T/NK cell sub-clusters: for example, T-reg cells (cluster 3–5) highly express immune checkpoint receptors CTLA4 and TIGIT and have strong immune suppressive role in PDAC microenvironment, while CD8^+^ cells show low expression of activation markers (granzymes, perforin, IFNγ) and high exhaustion markers (LAG3 and EOMES).

Single cell transcriptomic and related t-SNE analysis conducted over 962 fibroblasts confirmed the two already known CAF categories (iCAFs and myCAFs) [59,60], each of which with its unique gene signature (see Figure 3).

More in detail, iCAFs (fibroblast sub-cluster 1) show high expression of cytokines (IL-6, IL-8), chemokines (CXCL1, CXCL2, CCL2, CXCL12) and matrix protein genes (LMNA and DPT). These cells strongly express also HAS1 (hyaluronan synthase 1) and HAS2 (hyaluronan synthase 2) genes, which are the main responsible for hyaluronan synthesis, the principle component of extracellular matrix in pancreatic cancer. The majority of fibroblasts clustered within sub-cluster 2, which is composed of cells previously identified as myCAFs: They express ACTA2 (encoding α-SMA) and other contractile protein genes (TAGLN, MYL9, TMP1, TMP2, and MMP11).

A third CAF sub-cluster was identified through scRNA-seq of PDAC KPC mouse models and subsequently confirmed in human tissue samples: This cluster is clearly distinct from iCAFs and myCAFs, since it expressed class II MHC related genes (usually limited to “professional” antigen presenting cells), thus it was called “antigen presenting CAF” (apCAF). These cells strongly express pan-fibroblast markers—and this confirms their “fibroblastic nature”—and they upregulate several pathways, over all antigen presentation and processing, fatty acid metabolism and MTORC1 signaling. The Figure 3 summarize the main CAFs subpopulations in PDAC discovered so far.

Imaging mass cytometry (IMC) technology with several metal-conjugated antibodies confirmed the existence of the new CAF subtype in human PDAC tissues: These human apCAFs, as for the murine ones, mildly to highly express class II HLA genes (HLA-DRA, HLA-DPA1, HLA-DQA1) as well as CD74 and SLPI.

Considering the “unusual” expression of class II MHC genes on this sub-population, the authors hypothesize that apCAFs could interact with immune system, and especially with CD4^+^ T cells: This though was confirmed, as apCAFs induce CD25 and CD69 in co-cultured T cells (Figure 3). Therefore, this CAF population is supposed to be strongly involved in immune suppression, as it do not express co-stimulating molecules (like CD80, CD86, and CD40), necessary for T lymphocyte activation. In more detail, class II MHC expressed by apCAFs could act as a “decoy” receptor, because it inactivates T cells that are exposed to it (since there are no co-stimulating molecules), inducing anergy or T-reg differentiation.

Dominguez CX et al. [19] further expanded our knowledge about CAF subpopulations in PDAC. They initially confirmed the existence of putative iCAFs and myCAFs in pancreatic cancer KPP mouse models as well as the distinct molecular mechanisms (already showed by Biffi et al. [60]) of differentiation processes between these different CAF populations. Furthermore, they showed that the identified myCAF cluster (induced by TGF-β) closely surrounds the cancer islet (confirming data of Öhlund D et al. [59]), and highly express a molecule called LRRC15 (Leucine rich repeat containing 15). Of note, these LRRC15^+^ stromal cells could directly enhance tumor growth, forming a specific pro-tumor niche in mouse models. Moreover, reanalyzing human single cell data from Peng et al. [17], as well as other PDAC datasets, the authors discovered three main clusters of fibroblasts: The majority of cells (52%) surrounds tumor islet and expresses high levels of LRRC15 itself and ACTA2 (both related to TGF-β signaling), suggesting that myCAF is the prominent population in PDAC microenvironment (concordantly with Elyada et al. [18]). However, Dominguez CX et al. did not find apCAF signature in any stromal cell type, although all human CAFs identified seem to express HLA-DRA and CD74, as apCAF population already showed [18]. More importantly, and with translational relevance, the authors found high LRRC15 expression by non-malignant stromal elements in many other tumors (mainly breast and head and neck cancers): This “LRRC15 CAF signature” correlates with poor response to immune checkpoint blockade (for example anti-PD-1 Atezolizumab), due to strong immunosuppression properties linked to these cells.

The last classification we discuss [43] is also the most relevant, in terms of molecular and clinical implications, since the authors try to integrate genomics with bulk and single cell transcriptomic data. Chan-Seng-Yue M et al. analyzed very recently 330 cancer samples from 314 pancreatic cancer patients (stage I/II: 62%; locally advanced: 6%; stage IV: 31%) with WGS as well as with RNA-seq, although the latter one was performed on 248 samples only. The aim of the work is to re-define molecular subtypes of PDAC (almost 90% of analyzed samples): bulk RNA-seq classified 248 PDACs into five robust clusters (Table 3): Basal-like A, Basal-like B, Classical-A, Classical-B, and Hybrid tumors. These subtypes greatly differ each other in terms of molecular and clinical features and their distribution highly vary according to disease stage.

Classical A/B tumors predominate in early stages (62% of stage I/II versus 46% of stage IV), and the most represented is Classical-A subtype (44% of all patients stage I/II and 36% of all patients stage IV). A molecular hallmark of both classical types is a strong upregulation of transcription factors related to pancreatic lineage differentiation (GATA6, GATA4, HNF1A, HNF4G, ONECUT2, and others) as well as complete loss of SMAD4. Of note, only GATA6 is found to be recurrently amplified in the genome, although even tumors lacking amplification highly express GATA6 itself. As GATA6 is strongly expressed by normal pancreatic ducts, the authors hypothesize that “classical” phenotype is the default pathway of pancreatic carcinogenesis, also considering that classical subtypes are the most abundant clusters, regardless of clinical disease stage.

Basal-like subtypes are differently distributed among PDAC patients according to disease stage: Basal-like A tumors are very rare in early stage (5% of all stage I/II patients) but they constitute almost ¼ of all stage IV patients (24%). Conversely, Basal-like B cancers predominate in earlier stages (stage I/II/III). This shows that the overall poor prognosis of “Basal-like” signatures previously identified (quasi-mesenchymal [12], basal-like [13], and squamous [14]) depends in earlier stages on Basal-like B tumors, while in advanced stage is almost exclusively linked to chemo-resistant Basal-like A cancers. Molecular upregulated pathways by Basal-like A/B tumors are TP53 (with related gene network), epithelial-to-mesenchymal transition and TGF-β pathway: in detail, these cancers are highly enriched for TP53 mutations (78% of cases) and complete loss of CDKN2A (87% of cases). The main difference between the two identified Basal-like subtypes consists in the “squamous signature”: Basal-like A shows high squamous expression program (related to poor prognosis, see later) and it appears to be selected during disease progression.

To explore in more detail how expression signature of bulk RNA-seq distribute intra-tumorally, the authors performed scRNA-seq of PDAC samples from 15 patients (13 early stages and 2 stage IV). They analyzed 31.195 cells and performed an ultra-detailed analysis of tumor cells only, clustering them with Basal-like and Classical signatures. In the great majority of patients (13/15), they found both classical and basal-like cancer cell clusters within the same tumor at the same time, greatly underlying the importance of single cell analysis in cancer phenotype description. Moreover, this notion complicates things even more, because it has conclusively shown that bulk transcriptomic cannot resolve the great complexity of intratumoral heterogeneity.

Indeed, the different proportion of classic-like and basal-like cancer cells in different tumors creates a “transcriptional continuum” at the bulk RNAseq level, from which Hybrid subtype originates. Hybrid tumors predominate in early disease (24% of all stage I/II patients), but they are also well represented in metastatic disease too (18% of all stage IV patients).

More importantly, the authors try to understand how different transcriptional subtypes correlate to different survival outcomes in advanced stage patients. They analyze survival data from 80 stage III/IV PDAC patients treated with modified FOLFIRINOX (mFOLFIRINOX) [61] or a gemcitabine-based regimen (like gemcitabine plus nab-paclitaxel [62]): Basal-like A patients showed the worst prognosis ever in terms of overall survival, compared to all other subtypes (HR: 0.62, *p* = 0.12). Moreover, they analyze the objective response rates to chemotherapy of 66 advanced PDAC patients for whom data are available. Again, Basal-like A patients show the worst responses (regardless of the chemotherapy regimen used), as 70% of them develop progressive disease, while other subtypes are more or less uniformly distributed between partial responses or stable disease. Unfortunately, the authors could not infer a precise correlation between tumor subtypes and objective response to a specific chemotherapy regimen (mFOLFIRINOX versus gemcitabine-based) because of the very few patients analyzed, although it seems that Basal-like A could receive perhaps more benefit from gemcitabine-based regimens than from mFOLFIRINOX.

The same research group has very recently confirmed within the COMPASS trial [63] the role of GATA6 expression as a simple surrogate biomarker to define “classical” transcriptomic type, as well as the very high chemo-resistance nature of basal-like tumors [64]. Indeed, basal-like tumors treated with mFOLFIRINOX showed a higher progression rate (60%) compared to that of Classical PDACs (15%), *p* = 0.0002, suggesting once more that perhaps mFOLFIRINOX should not be the right chemotherapy regimen for Basal-like cancers.

However, to date there is no molecular selection available for advanced PDAC patients beginning first line chemotherapy: Both mFOLFIRINOX [61] and Gemcitabine plus nab-paclitaxel [62] should be considered correct choices, assuming clinical factors like stage (locally advanced versus stage IV), age and general clinical conditions. Only perspective molecular-guided clinical trials like COMPASS [63] will definitively clarify subtype-specific chemotherapy regimen to use, opening even for this malignancy a new era of “precision medicine”.

## 6. Conclusions and Future Perspectives

The genomic and transcriptomic (bulk and single cell) classifications of pancreatic cancer reviewed so far have shown a highly heterogeneous and enormously complicated disease picture, with many malignant, stromal and immune subpopulations shaping the complexity of tumor microenvironment.

It is now a matter of fact that every single pancreatic tumor should be considered as a mixture of many different cancer cells, some with classic-like and others with basal-like signature [43,65], although at the bulk level they could be recognized as only “classical” or “basal-like” type. Moreover, a specific identified transcriptomic cluster for a given tumor should not be conceived as a “crystallized molecular subtype”, not susceptible to change over time or with therapy. Indeed Chan-Seng-Yue M et al. [43] showed that “subtype switch” (from Basal-like to Classical type and vice versa) during therapy is a reality, probably due to the expansion of small previously unidentified sub-clones, and this could complicate even more the therapeutic management of PDAC patients.

In fact, preliminary evidences suggest that classic-type PDACs respond to mFOLFIRINOX in advanced stage [43,64], as well as they seem to receive benefit from adjuvant 5-FU if resected [45], while basal-like tumors (mainly basal-like A) are highly resistant to 5-FU based chemotherapy regimens, and we should consider an alternative drug combination (like a gemcitabine-based one). However, even if we use from the beginning the suggested right regimen for a specific tumor type, “subtype-switch” unavoidably could occur and could be responsible for the disease progression.

Indeed, intra-tumor heterogeneity remains the strongest barrier to cancer therapy as well as the main cause of drug resistance: only the power of single cell analysis could lead to a complete dissection of all tumoral cell states, even the smallest dangerous sub-clones, identifying their specific vulnerabilities. For example, Byers et al. [66] recently performed sc-RNAseq of circulating tumor cells (CTC) derived xenografts from small cell lung cancer patients: They showed a dramatic increasing of intra-tumor heterogeneity and epithelial-to-mesenchymal transition in cellular populations derived from platinum-resistant patients, underlining that treatment resistance could rise from coexistence of multiple subpopulations of cell with highly heterogeneous gene expression programs. Moreover, although DNA bulk sequencing can greatly inform cancer biology, it cannot distinguish which mutations occur in the same clone(s) as well as it cannot measure in deep clonal complexity. Single-cell DNA sequencing (sc-DNAseq) could overcome this issue, especially if associated to sc-RNAseq: preliminary and unpublished evidences in acute myeloid leukemia [67] suggest that scDNA-seq could precisely map the clonal trajectory of each patient, identifying the specific mutation combination that synergize to promote clonal expansion and dominance. Therefore, a combination of single cell genomic and transcriptomic tools applied to pancreatic cancer is highly desirable, and probably the only way to reveal the complete PDAC sub-clonal structure.

In addition to cancer cell plasticity, also microenvironment is a strong barrier to therapy, as sc-RNAseq has revealed countless stromal and immune cell sub-clusters with different pro-tumoral properties.

Among others, TGF-β induced LRRC15^+^ myofibroblasts strongly stimulate the survival of adjacent cancer cells, and can reprogram the tumor microenvironment in pro-tumor sense. To complicate picture even more, regulatory T cells were recently identified [68] as a key source (in addition to cancer cells) of TGF-β itself in PanIN, but their depletion in mouse models, although associated to a significant reduction of myCAFs, leads to a strong increase of MDSCs recruitment (through CCR1 receptor), with an additional shift towards immune suppressive microenvironment. Bernard V et al. [16] have already shown that the majority of single profiled immune cells within PDAC microenvironment were MDSCs, and the work by Zhang [68] points out a possible molecular mechanism of their recruitment, and it hypothesizes a MDSCs-direct therapy with CCR1 inhibition, to reverse their “negative” pro-tumor impact on cancer microenvironment.

As cancer cells do, also CAFs are extremely dynamic, because they could easily differentiate in culture and in vivo from one subtype to another, with different biological properties. Therefore, it is possible to imagine in the future a “CAF-tailored” therapy, which is a therapy able to induce depletion of a particular CAF cell state (for example a pro-tumoral one) and enrichment in a “positive” anti-tumoral one. More in detail, Djurec M et al. [69] recently showed that murine Saa3^+^ (Serum Amyloid A-3) CAFs (a population probably related to murine and human iCAFs [18]) highly stimulate PDAC tumor cells growth in orthotopic models: concordantly, the knockdown of Saa3 in these CAFs is able to reprogram their pro-tumoral functions, inhibiting tumor growth itself. Since SAA1 (Serum Amyloid A1), the human ortholog of murine Saa3, is overexpressed in human PDAC CAFs, it is possible to imagine a “reprogramming” therapy against this CAF population, as we previously mentioned. However, we are still far away from such a reality, because we do not know yet the entire picture of microenvironmental interactions between the various immune and stromal components each other and with cancer cells. Moreover, it is very likely that the CAF subpopulations discovered so far are only the tip of the iceberg, and many more CAF cell states will be discovered over the next years, as we greatly expand the absolute number of PDAC single cells sequenced.

Indeed, the next years will probably see an even bigger explosion of single cell papers characterizing PDAC, as the Human Cell Atlas (HCA) project [70] and Human Biomolecular Atlas Program (HuBMAP) [71] were already launched, with the aim of dissect every single cell of our body (in health and in disease) using single cell genomic tools. The first papers from HCA were already published [72,73], and they greatly expanded our previous knowledge of maternal–fetal interface [72] and thymus [73], with novel cellular subpopulation described for the first time.

Moreover, the precise genomic and transcriptomic dissection of tumor-derived organoids is now a reality, and it could lead to a more precise drug assignment for each PDAC patient. Indeed Tuveson et al. [74] showed that patient-derived organoids precisely recapitulate the genomic and transcriptional subtypes of primary pancreatic tumors, helping us to establish patient sensitivity or resistance to different therapeutic strategies, both in the adjuvant setting as well as in advanced stage. Bulk and, especially, single cell analysis of patient-derived organoids will lead to a more precise disease taxonomy, hopefully with important clinical implications.

Another hope for precision medicine in pancreatic cancer resides in liquid biopsy [75], a non-invasive approach (blood sampling based) that could identify circulating tumor DNA (ctDNA) or circulating tumor cells (CTCs), with relevant implications. First experiences speculated a role for ctDNA (KRAS gene mutations) in early disease diagnosis: in the study by Lennon AM et al. [76] liquid biopsy specificity was high (99.5%), although at the prize of lower sensitivity (30% to 64%), meaning that in some patients tumor DNA could not be identified (probably due to overall low levels of DNA or lacking of KRAS mutations). In 2019, Gibbs P et al. [77] investigated ctDNA as a surrogate marker of benefit from gemcitabine adjuvant chemotherapy in resected PDAC patients: They showed that detectable ctDNA following curative intent was associated to poor prognosis and disease relapse, despite the adjuvant chemotherapy performed; this is a precious information but requiring validation in clinical trials. Nowadays, liquid biopsy experiences to guide “in real time” PDAC therapy in advanced disease are lacking, but they are highly desirable, especially if combined to single cell genomic tools to analyze genome/transcriptome of single circulating cells.

In conclusion, we strongly believe that only through the complete molecular and cellular dissection of hundreds of PDAC samples (possibly primary and metastasis paired as well as single circulating cells), we will build a comprehensive atlas of the disease, with the subsequent ability to design novel microenvironmental- and cancer-directed therapeutic strategies for this still highly lethal disease.

## Figures and Tables

**Figure 1 ijms-21-02814-f001:**
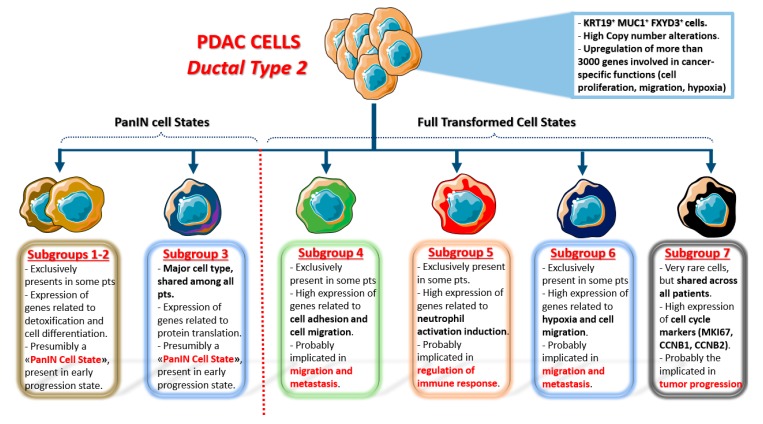
Single cell sub-clusters of malignant ductal cells as emerged by Peng et al. [17]. Created with Servier Medical Art.

**Figure 2 ijms-21-02814-f002:**
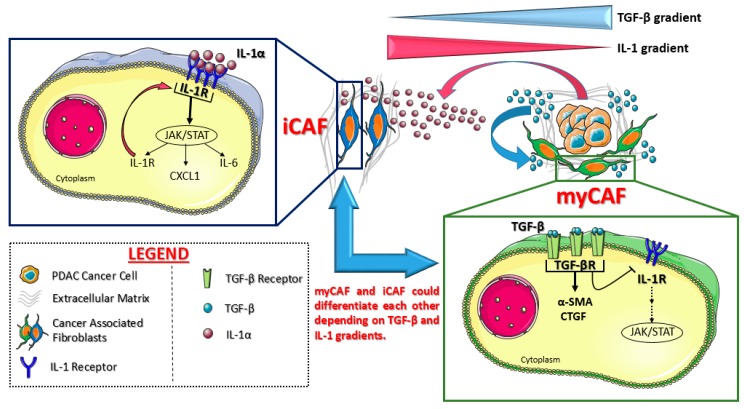
Cancer-associated fibroblast (CAF) subtypes identified in pancreatic cancer by Öhlund D et al. [59] and Biffi et al. [60]. Myofibroblasts (myCAFs) reside in close contact with tumor niches: their phenotype depends on TGF-β signaling, which inhibits IL-1 molecular cascade. On the other side, IL-1 reprograms the CAFs far away from tumor cells into inflammatory CAFs (iCAFs), with a positive feedback loop on IL-1 receptor itself. These two CAF populations appear plastic and interchangeable, depending on soluble factor gradients. Created with Servier Medical Art.

**Figure 3 ijms-21-02814-f003:**
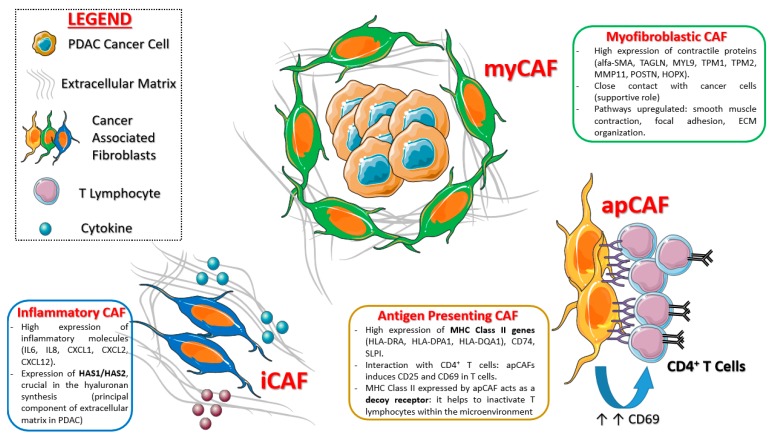
Main CAF sub-types and related functions postulated in pancreatic cancer. Created with Servier Medical Art.

**Table 1 ijms-21-02814-t001:** Comparison between the core pathways identified in pancreatic ductal adenocarcinoma (PDAC) by Jones et al. [9] and Bailey et al. [14] with related frequencies of mutation.

GENOMIC STUDIES							
Jones S. et al. [9] *Patients = 24*				Bailey P. et al. [14] *Patients = 456*			
CORE PATHWAYS			Mutation Frequencies (%)	CORE PATHWAYS			Mutation Frequencies (%)
PATHWAYS		GENES		PATHWAYS		GENES	
**1**	**RAS-MAPK Pathway**	**KRAS**, MAP2K4, RASGRP3	**100%**	**1**	**RAS-MAPK Pathway**	**KRAS**, MAPK4	**92%**
**2**	**Cell Cycle**	**CDKN2A**, APC2, FBXW7	**100%**	**2**	**Cell Cycle/DNA Damage Control**	**TP53**, **CDKN2A**, TP53BP2	**78%**
**3**	**DNA Damage control**	**TP53**, RANBP2, EP300, ERCC4/6	**83%**	**3**	**DNA Repair**	BRCA1, BRCA2, ATM, PALB2, ATF2	**12% Somatic** **5% Germline**
**4**	**Apoptosis**	CASP10, VCP, CAD, HIP1	**100%**				
**5**	**TGF-Beta Pathway**	**SMAD4**, SMAD3	**100%**	**4**	**TGF-Beta Pathway**	**SMAD4**, SMAD3, TGFBR1, TGFBR2, ACVR1B, ACVR2A	**47%**
**6**	**WNT/Notch Pathway**	MYC, WNT9A, MAP2, TSC2, GATA6, TCF4	**100%**	**5**	**WNT Pathway**	RNF43, MARK2, TLE4	**5%**
				**6**	**Notch Pathway**	JAG1, NF2, BCORL1, FBXW7	**NR**
**7**	**Hedgehog Pathway**	GLI1, GLI3, BOC, CREBBP, LRP2	**100%**	**7**	**ROBO/SLIT Pathway**	ROBO1, ROBO2, SLIT, MYCBP2	**NR**
**8**	**JNK pathway**	TNF, ATF2, NFATC3, MAP4K3	**96%**	**8**	**RNA Processing**	SF3B1, U2AF1, REM10	**16%**
**9**	**“Invasion” pathway**	ADAM11/12/19, DPP6, MEP1A	**92%**	**9**	**Chromatin Remodeling**	KDM6A, MLL2, MLL3, SET2D	**24%**
**10**	**Cell adhesion**	CDH1, CDH10, CDH2, CDH7	**79%**	**10**	**SWI/SNF Pathway**	ARID1A, ARID1B, SMARCA4, PERM1	**14%**
**11**	**Small GPTase pathway**	CDC42BPA, AGHGEF7, ARHGEF9	**79%**				
**12**	**Integrin signaling**	ITGA4, ITGA9, ITGA11, LAMA1	**67%**				

**Table 2 ijms-21-02814-t002:** Comprehensive list of the most frequent mutated genes (grouped by different cellular processes) in PDACs across various datasets. The table also provides clinical and pathological informations.

GENOMIC STUDIES					
PAPERS		Jones S et al. [9]	Waddell N et al. [10]	TCGA [11]	Connor AA et al. [33]
***Patients/Samples***		*Patients = 24*	*Patients = 100*	*Patients = 150*	*Patients = 289*
		*Samples = 24*	*Samples = 100*	*Samples = 149*	*Samples = 319*
***Stage***		*Advanced (ranging IIB–IV)*	*Early to Advanced (ranging IA to IV)*	*Early to Advanced (ranging I to IV)*	*Early to Advanced (ranging I to IV)*
***Histology***		*PDAC*	*PDAC*	*PDAC*	*PDAC*
***Grading***		*NR*	*G1: 2%* *G2: 57%* *G3: 33%* *NR: 7%*	*G1: 3%* *G2: 50%* *G3: 47%*	*NR*
***Microdissection***		*No*	*No*	*No*	*Yes*
**Mutated Genes**		**FREQUENCIES (%)**			
**RAS-MAPK Pathway**	**KRAS**	**100%**	**95%**	**93% ^*^**	**89%**
	**BRAF**	**NR**	**1%**	**3% ^*^**	**3%**
**TGF-Beta** **Pathway**	**SMAD4**	**33%**	**31% ^*^**	**32% ^*^**	**25%**
	**TGFBR2**	**13%**	**8%**	**5% ^*^**	**3%**
**DNA Damage Response**	**CDKN2A**	**8%**	**35% ^*^**	**30% ^*^**	**26%**
	**TP53**	**75%**	**74% ^*^**	**72% ^*^**	**80%**
**Chromatin** **Remodeling**	**KDM6A**	**NR**	**7%**	**3% ^*^**	**5%**
	**MLL3**	**17%**	**13%**	**4% ^*^**	**NR**
	**ARID1A**	**8%**	**16%**	**6% ^*^**	**9%**
**DNA Repair**	**ATM**	**0%**	**3%**	**5% ^*^**	**NR**
	**BRCA1**	**NR**	**4%**	**1% ^*^**	**NR**
	**BRCA2**	**NR**	**3%**	**4% ^*^**	**NR**
	**PALB2**	**NR**	**1%**	**1% ^*^**	**NR**
**Axon** **Guidance** **Pathway**	**ROBO2**	**0%**	**3%**	**NR**	**NR**
	**SLIT2**	**0%**	**4%**	**NR**	**NR**
	**SEMA5A**	**0%**	**1%**	**NR**	**NR**
	**SEMA5B**	**8%**	**1%**	**NR**	**NR**
**Pancreatic Differentiation**	**GATA6**	**NR**	**3%**	**9% ^*^**	**NR**
**RNA Processing**	**SF3B1**	**8%**	**4%**	**NR**	**NR**
**WNT Signaling**	**RNF43**	**NR**	**8%**	**7% ^*^**	**5%**

**^*^** These data refers to deleterious point mutations plus structural variation (amplifications/deletions). *NR = not reported.*

**Table 3 ijms-21-02814-t003:** Main bulk transcriptomic subtypes of pancreatic cancers. For each identified subtype, informations related to putative cell markers, pharmacological sensitivity (if available) and prognosis are provided.

TRANSCRIPTOMIC STUDIES	TRANSCRIPTOMIC SUBTYPE				
	*TUMORAL COMPONENT RELATED*			*STROMA RELATED*	
	“CLASSICAL”	“BASAL LIKE”	Other Types		
Collisson et al. [12]	**Classical Type**	**Quasi-Mesenchymal Type**	**Exocrine-Like Type**	**-----**	
**Putative Markers**	GATA6^+^	GATA6^-^	ELA3A^+^ CTFR ^+^		
**Pharmacological sensitivity**	↑ Erlotinib ↓ Gemcitabine	↓ Erlotinib ↑ Gemcitabine	NR		
**Prognosis**	Good	Poor	NR		
Moffitt et al. [13]	**Classical Type**	**Basal-like Type**	**-----**	**Normal Stroma**	**Activated Stroma**
**Putative Markers**	GATA6^+^	GATA6^-^		Pancreatic stellate cell markers	Macrophage markers
**Prognosis**	Good	Poor		Good	Poor
Bailey et al. [14]	**Pancreatic Progenitor**	**Squamous Type**	**ADEX Type**	**Immunogenic Type**	
**Putative Markers**	Apomucins^+^ PDX1^+^, FOXA2/A3^+^	TP63^+^, GATA6^-^	NR5A2^+^, MIST1^+^	Significant immune infiltrate (CD4^+^/CD8^+^ T cells), CTLA4^+^, PD1^+^	
**Pharmacological sensitivity**	NR	NR	NR	Supposed response to immune checkpoint inhibitors	
Prognosis	Good	Poor	Good	Good	
Puleo et al. [15]	**Pure Classic/Immune Classic**	**Pure Basal-Like**	**-----**	**Desmoplastic**	**Stroma Activated**
**Putative Markers**	GATA6^+^	GATA6^-^		Structural stromal component	ACTA2^+^, FAP^+^
**Prognosis**	Good	Poor		Intermediate	Intermediate
Chan-Seng-Yue et al. [43]	**Classical A /** **Classical B**	**Basal-like A /** **Basal-like B**	**Hybrid**	**-----**	
**Putative Markers**	GATA6^+^, GATA4^+^	GATA6^-^, Squamous signature (type A)	GATA6^+/-^		
**Pharmacological sensitivity**	↑ mFOLFIRINOX ↓ Gemcitabine-based	↑ Gemcitabine based ↓ mFOLFIRINOX	NR		
**Prognosis**	Good	Poor	Intermediate		

NR = not reported. **-----** indicates that for the present category (ex “stroma related” for “Collisson et al.”), there are not these subtypes.

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
