# Peer review of "Pancreatic Cancer Molecular Classifications: From Bulk Genomics to Single Cell Analysis"

_ijms, 2020, doi:10.3390/ijms21082814_

Round 1

Reviewer 1 Report

A very nice review about the genomic developments of pancreatic cancer based on the recent publications. A major caveat about table 2. The patients in the different studies are not independent, meaning the authors shouls only compare one manuscript of the Biankin group and one of the TCGA group, since smaller studies might have been integrated in larger studies partly leading to patients analysed more than one time.

Author Response

Point 1: A very nice review about the genomic developments of pancreatic cancer based on the recent publications. A major caveat about table 2. The patients in the different studies are not independent, meaning the authors shouls only compare one manuscript of the Biankin group and one of the TCGA group, since smaller studies might have been integrated in larger studies partly leading to patients analysed more than one time.

Response 1: We tried to resolve the issue you noted, by eliminating from the table 2 (pages 6 – 7) the part related to Biankin et al: in this way only one study remains in the table from the Biankin / Grimmond research group, that is ref. 10. Moreover, we modified the table itself, according to Reviewer 2, to include also data regarding to clinical informations (stage, histology, grading, microdissection): in fact, these could be useful for the reader.

We thank you again for this precious indication, to improve even more the quality of our paper.

Reviewer 2 Report

Pompella and collaborators have prepared a very comprehensive paper that summarize very well the studies in the pancreatic cancer field, pointing to future of the studies to establish the PDAC subtypes with the final goal of achieving precision medicine for PDAC patients.

Before publication the authors should try to address the following issues:

  1. Table 1. There are genes classified under different pathways in both papers. For instance, p53 DNA damage vs cell cycle.

  1. Table 2. Bailey study should also be included on Table 2

  1. Table 2: please, define the differences between patients, samples and tumor grade. And from the samples, please, indicate the ones that are studied after microdissection.

  1. Maybe it should be pointed or discussed that differences between all the studies on Table 2 can also reflect differences in the histological data of tumors from the different studies.

  1. On table 2, the reference 28, is the 29 (Biankin et al)

  1. On line 212 is Waddell, not Weddell

  1. On line 214, I guess that the authors meant MYC

  1. Table 3: include some markers genes for each classification, at least for the most important ones. Drug sensitivity should also be mention, in this or in another table.

  1. Figure 2: interconversion between CAFs should be included in this or in another figure.

  1. Authors mention on line 538 or in the conclusion (line 716) the therapeutic potential of targeting the stroma. They may consider to include the “Djurec et al study (PNAS 2018 doi:10.1073/pnas.1717802115)” that illustrates the pattern of expression of PDGFRa+ CAFs and normal pancreatic fibroblasts. Importantly, this study shows that elimination in CAFs of Saa3 (SAA1 in human), gene expressed in the iCAFs and the highest upregulated gene in this subset of CAFs, reprogram the CAFs, inhibiting their protumoral characteristics.

  1. Please, include in a table or in a figure the information about the sensitivity to different treatments.

  1. Consider to include in the Conclusion section that generation of Organoids from different PDAC subtypes may help to stablish their sensitivity to different therapeutic strategies.

Author Response

Point 1: Table 1. There are genes classified under different pathways in both papers. For instance, p53 DNA damage vs cell cycle.

Response 1: In Table 1 (page 4) we corrected the “TP53 section” for Bailey et al (we added in this section also “DNA Damage Control”).

Point 2: Table 2. Bailey study should also be included on Table 2

Point 3: Table 2: please, define the differences between patients, samples and tumor grade. And from the samples, please, indicate the ones that are studied after microdissection.

Responses 2 and 3: we modified Table 2 (pages 6-7), according to your kind suggestion, to include also data regarding clinical informations (stage, histology, grading, microdissection). We do not choose to add Bailey et al data [14] in the table, because patients from Waddell et al [10] and patients from Bailey et al [14] are not independent (some patients were analyzed more than one time in these two datasets). For the same reason we choose to remove the Biankin data [32] from the table, according also to Reviewer 1 comment. In this way, only Waddell et al data [10] from the Grimmond/Biankin group is present in the table 2 itself.

Point 4: Maybe it should be pointed or discussed that differences between all the studies on Table 2 can also reflect differences in the histological data of tumors from the different studies.

Response 4: we added a sentence at the end of section 3 (“Pancreatic Ductal Adenocarcinoma: from histology to early genomic studies”), line 290-295.

Point 5: On table 2, the reference 28, is the 29 (Biankin et al)

Response 5: we removed Biankin data from this table, as described above (points two-three).

Point 6: On line 212 is Waddell, not Weddell.

Response 6: we corrected this error (now line 222).

Point 7: On line 214, I guess that the authors meant MYC

Response 7: we corrected this error (now line 224).

Point 8: Table 3: include some markers genes for each classification, at least for the most important ones. Drug sensitivity should also be mention, in this or in another table.

Response 8: we completely modified table 3 (page 11) according to your kind suggestion. Now the table reports informations about main putative markers, prognosis and drug responses (point eleven you formulated) for the identified transcriptomic subtypes.

Point 9: Figure 2: interconversion between CAFs should be included in this or in another figure.

Response 9: we totally agree with your suggestion. Indeed, we created a completely new figure (now it is the “new” figure 2 while the “old figure 2” is now figure 3) in which we showed the relationship between iCAFs and myCAFs within the tumor microenvironment, as well as their possible interconversion. This Figure 2 is now on page 17.

Point 10: Authors mention on line 538 or in the conclusion (line 716) the therapeutic potential of targeting the stroma. They may consider to include the “Djurec et al study (PNAS 2018 doi:10.1073/pnas.1717802115)” that illustrates the pattern of expression of PDGFRa+ CAFs and normal pancreatic fibroblasts. Importantly, this study shows that elimination in CAFs of Saa3 (SAA1 in human), gene expressed in the iCAFs and the highest upregulated gene in this subset of CAFs, reprogram the CAFs, inhibiting their protumoral characteristics.

Response 10: absolutely agree with you. Indeed, we cited this outstanding study in section 6 of the paper (“Conclusion and future perspectives”), and we discussed it (line 776-781). In this way, we could explain more effectively the concept of “reprogramming therapy” of cancer associated fibroblasts.

Point 11: Please, include in a table or in a figure the information about the sensitivity to different treatments.

Response 11: as already mentioned in response 8, we we completely modified table 3 (page 11). Now the table reports informations about main putative markers, prognosis and drug responses (point eleven you formulated) for the identified transcriptomic subtypes.

Point 12: Consider to include in the Conclusion section that generation of Organoids from different PDAC subtypes may help to stablish their sensitivity to different therapeutic strategies.

Response 12: we strongly agree with you about the importance of tumor derived organoid profiling in PDAC. We added in the conclusion section of our paper this fundamental issue (line 793-799).

We thank you again for these precious indications, to improve even more the quality of our paper.

Reviewer 3 Report

This is a well-Written, nicely illustrated and timely review on pancreatic cancer. The authors should be congratulated for their work also using ad hoc tables that are very useful for the readers.

Here below some suggestions to improve this paper:

  • a brief review on the oncogenesis of pancreatic Cancer through its precursors, with a specific focus on molecular cascade, is needed (suggested references PMID are: 28188630, 24604757, 30364837);
  • a better discussion on the rare but interesting KRAS wild-type cases should include also Kinase fusion genes
  • the contribution of single cell sequencing should be mentioned also in terms of a better comprehension of tumor heterogeneity (aBove all as a future perspective)
  • the role of liquid biopsy for molecular profiling of pancreatic Cancer should be commented (suggested references PMID are: 28874546, 27433079, 31405192).

Author Response

Point 1: a brief review on the oncogenesis of pancreatic Cancer through its precursors, with a specific focus on molecular cascade, is needed (suggested references PMID are: 28188630, 24604757, 30364837);

Response 1: we totally agree with you about the need of more expanded discussion of PanIN and IPMN molecular cascades that lead to overt PDAC. Therefore, we added in the section 2 (“Pancreatic ductal adenocarcinoma: from histology to early genomic studies”) a discussion about this crucial topic, line 86 – 97, including the three citations you suggested.

Point 2: a better discussion on the rare but interesting KRAS wild-type cases should include also Kinase fusion genes

Response 2: again we strongly agree with you about the relevance of this topic (in the light of targeted therapy possibilities for the KRAS WT patients with kinase fusion genes). Therefore, we added, as kindly suggested, more informations about KRAS WT pancreatic tumors, with emphasis on kinase fusion genes within the section 3 “Genome Sequencing of PDAC: from exome sequencing to whole genomes and multi-omics”, line 232 - 241.

Point 3: the contribution of single cell sequencing should be mentioned also in terms of a better comprehension of tumor heterogeneity (aBove all as a future perspective).

Response 3: We added a more in deep discussion about intra-tumor heterogeneity and single cell sequencing in the “conclusion and future perspectives” section, with new references, line 743 - 758.

Point 4: the role of liquid biopsy for molecular profiling of pancreatic Cancer should be commented (suggested references PMID are: 28874546, 27433079, 31405192).

Response 4: again we totally agree with you, regarding the importance of liquid biopsy topic. Therefore, we added a discussion about liquid biopsy at the end of the “conclusion and future perspective” section, line 800 – 812, integrating it with the rest of discussion, and using references you suggested.

We thank you again for these precious indications, to improve even more the quality of our paper.

Round 2

Reviewer 1 Report

Very nice review.